# Bearing defect detection based on the improved YOLOv5 algorithm

**Kangning Li, Peigang Jiao◯\*, Jiaming Ding, Weibo DU**

School of Construction Machinery, Shandong Jiaotong University, Jinan, Shandong Province, China

\* jiaopeigang@163.com

**Data Availability Statement:** All dataset files are available from the figshare database(https://doi.org/10.6084/m9.figshare.25335076.v1; https://doi.org/10.6084/m9.figshare.26197928.v1; https://doi.org/10.6084/m9.figshare.26421013.v1).

## Abstract

In the field of bearing defect detection, Aiming at the problem of low efficiency in manual inspection and prone to missed detections in scenarios with small target defects, and overlapping targets, an improved YOLOv5-based object detection method is proposed. Firstly, in terms of feature extraction, the C3 modules in the original backbone of YOLOv5 are replaced with the finer-grained Res2Block modules to improve the model's feature extraction ability. Secondly, in terms of feature fusion, a Bidirectional Feature Pyramid Network (BiFPN) is added to the original neck of YOLOv5 to enhance the fusion ability of shallow graphic features and deep semantic features. Finally, the performance of the improved YOLOv5 algorithm is validated through ablation experiments and comparative experiments with other defect detection algorithms, including the Small_obj algorithm the existing method of adding a small target detection head for identifying small target defects. The experimental results demonstrate that the improved YOLOv5 algorithm exhibits high mAP and accuracy in bearing defect detection, enabling more precise identification the types of small target defects on bearings in complex scenarios with multiple coexisting defects and overlapping detection targets, thereby providing valuable reference for practical bearing defect detection.

## Introduction

Bearings can be divided into rolling bearings and sliding bearings. As one of the most commonly used basic components of mechanical equipment worldwide, their main role is to support and transmit forces [1]. The quality of bearings directly affects the condition of equipment operation. Throughout the manufacturing process, bearings are prone to incur minor damages, including scratches and abrasions. In the prevailing practices within bearing factories, the identification of such defects primarily depends on human inspection. Given the subtlety of some of these damages, they can be challenging for inspectors to identify, consequently elevating the likelihood of both false detections and oversights in manual inspection processes. Hence, employing deep learning algorithms for accurately detecting bearing defects is of paramount importance.

Conventional defect detection techniques predominantly depend on the manual identification and analysis of defects. Chen Shuo [2] proposed a method to detect bearing defects using

**Funding:** This work was supported in part by the Shandong Provincial Department of Science and Technology Research Project on Key Technologies for the Development of All-Terrain Intelligent Orchard Platform. (2019GNC106032). The funder plays a role in the decision to publish this paper.

the end face of the bearing ring. Initially, this approach involves utilizing images of the bearing's end face to pinpoint the region of the ring's end face, followed by the application of the least squares method for contour fitting of the bearing end face, thereby identifying shape defects in the bearing. The defect type is subsequently determined based on its distinct features. Although this method has demonstrated a high overall detection rate of 98.6% in practical tests, it exhibits limited accuracy in scenarios involving bearing overlap and occlusion [3]. Duan Zhida and colleagues [4] have applied machine vision technology for the identification of defects in bearing rollers. They first enhance the contrast of the images, then utilize the Hough algorithm to extract targeted specific areas, and finally conduct edge template matching on the images to determine if there are any defects in the bearing rollers. The experimental results show that the accuracy of this method is higher than traditional predictive matching difference threshold methods and Gaussian weighted average segmentation methods, but its robustness in facing complex scenes is not very good. Taha, Z, and others [5] used ANN to collect and analyze the AE signals of bearings under test conditions. They trained on an AE signal dataset that was labeled based on whether defects were present or not, learning to associate specific patterns in the acoustic data with the corresponding bearing conditions.

As deep learning has advanced, a plethora of target detection algorithms grounded in Convolutional Neural Networks (CNN) has emerged, essentially bifurcated into two distinct classes. The first encompasses two-stage target detection algorithms, exemplified by R-CNN, SPP-Net, Fast R-CNN, Faster R-CNN, and R-FCN. These algorithms focus on finding the locations of target objects in the first stage, obtaining proposal boxes to ensure sufficient accuracy and recall rate. In the second stage, they focus on classifying the proposal boxes to find more precise locations [6]. Therefore, this type of algorithm usually has higher precision but is relatively slow in speed, unable to meet the needs of real-time detection. The second class is the one-stage target detection algorithms represented by the SSD series and YOLO series [7–10]. This type of algorithm does not need to obtain proposal boxes in the first stage of target detection and can directly produce the category probability and location coordinates of the object, directly obtaining the final detection result after a single detection [11]. This kind of algorithm has a faster operation speed than the two-stage algorithm, but the precision is somewhat reduced. PM, Bhatt, and others [12] classified literature on surface defect detection using deep learning technology, pointing out the challenges of traditional image processing technology in dealing with noise, changes in lighting conditions, and complex texture backgrounds, and proposed future research directions based on trends in the field of deep learning. Shenfield, A, and colleagues [13] proposed a dual-path neural network model that combines Recurrent Neural Networks (RNN) and CNN for diagnosing rolling bearing faults from raw data. Compared to existing methods, it demonstrates superior performance in domain adaptation and noise suppression tasks. Xu Qiang [14] improved the detection of steel surface defects by adopting a more lightweight network structure in place of the dense connection structure of the YOLOv3 algorithm, and by using a parallel transmission structure to further reduce the model's parameters. Additionally, the introduction of dilated convolutions enhanced the detection performance. While this method has increased the real-time capability of detection, the lightweight nature of the model has led to relatively lower accuracy in identifying small defects on the steel surface. Shi Zhenhua [15] use of an improved YOLOv3 feature fusion method can significantly reduce the number of redundant candidate boxes. This approach has high accuracy in identifying defects of single targets. However, the accuracy decreases in situations with multiple defect targets or where small target defects are dense.

To address the issues of low feature extraction performance and insufficient feature fusion in bearing defect detection algorithms, as well as the low detection and recognition rate of multiple types of defects in the bearing defect detection process, An improved algorithm based

on YOLOv5 has been proposed, aimed at improving the detection performance of bearing defects in complex scenarios. The main contributions of this paper are as follows:

1. In terms of feature extraction, to increase the receptive field of the improved YOLOv5 algorithm for small target defects in bearings, we have replaced the C3 module in YOLOv5 with a more finely grained Res2Block feature extraction module. This allows for better capture of detail information in images and improves the ability to recognize small target defects in bearings.

2. In terms of feature fusion, to fully utilize the image information on the dataset and enhance the feature fusion capability, the feature fusion network has been restructured with the addition of the BiFPN module. The BiFPN module enables more effective fusion of shallow image features with deep semantic features, which helps to reduce the miss rate and identification accuracy of bearing defects.

3. A performance comparison and detection result comparison were conducted with the Small_obj algorithm. The results show that the improved YOLOv5 algorithm presented in this paper increased the All mAP (average mAP across the dataset) by 4.9 percentage points and was able to detect bearing defects and identify bearing defect types more accurately.

## Related work

### Overall architecture of the model

The YOLOv5 series primarily consists of five algorithm types: v5s, v5m, v5l, v5x, and v5n. Among these, v5s offers the fastest detection speed but has lower accuracy. V5m provides a better balance between speed and accuracy. V5l has higher accuracy but slower detection speed. V5x boasts the best accuracy but has the slowest detection speed. V5n is a variant in the YOLOv5 series optimized for Nano devices, offering accuracy suitable for edge devices. Given the high real-time requirements of this paper, YOLOv5s is chosen as the basic model for research. The Small_obj algorithm model structure is shown in Fig 1, while the improved YOLOv5 algorithm model structure is shown in Fig 2.

As illustrated by the algorithm model structure in Fig 1, compared to the basic YOLOv5s algorithm model structure, this algorithm adds an additional detection head in the prediction part specifically for detecting small object defects, increasing the number of detection heads from three to four. At the same time, corresponding detection layers are added in the Backbone and Neck parts. This approach can improve the model's overall detection accuracy and stability, as well as its ability to recognize small object defects.

As shown by the algorithm model structure in Fig 2, the model mainly consists of four parts: input, backbone, neck, and prediction. The feature extraction network of the basic YOLOv5s algorithm utilizes the CSPDarknet53 module. This module, during the backward learning process, is prone to causing gradient repetition, which can reduce the model's learning capacity. To address this problem, the Res2Block module from Res2Net [16] is used to construct a new feature extraction network model. The Res2Block module offers the following significant advantages:

1. Reducing gradient repetition: The Res2Block module, by reorganizing the connections within the feature map, effectively reduces the issue of gradient repetition. This means that during back-propagation, the model is more likely to converge to a good solution, enhancing the model's training stability.

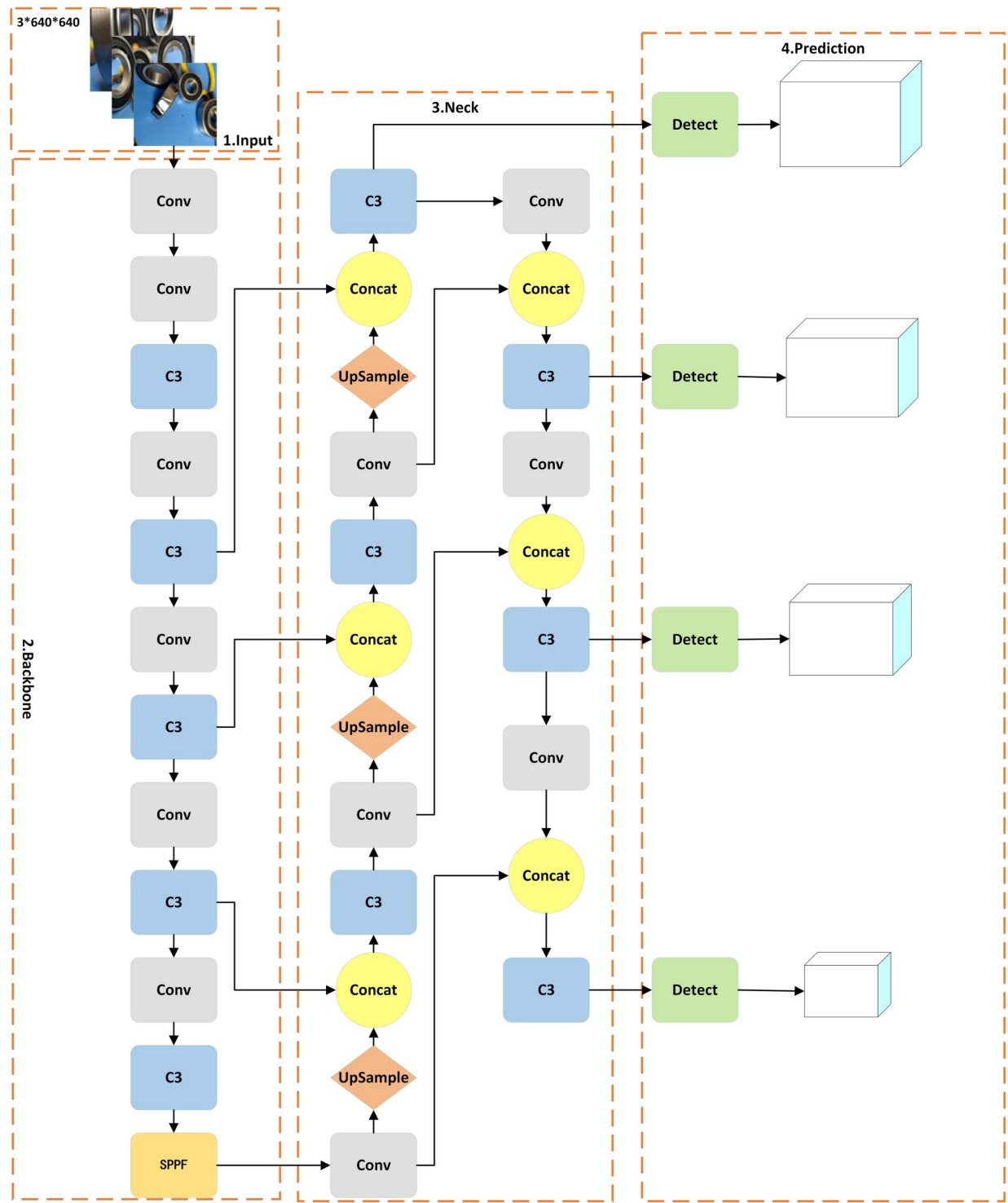

**Fig 1. Model structure of the Small_obj algorithm.**

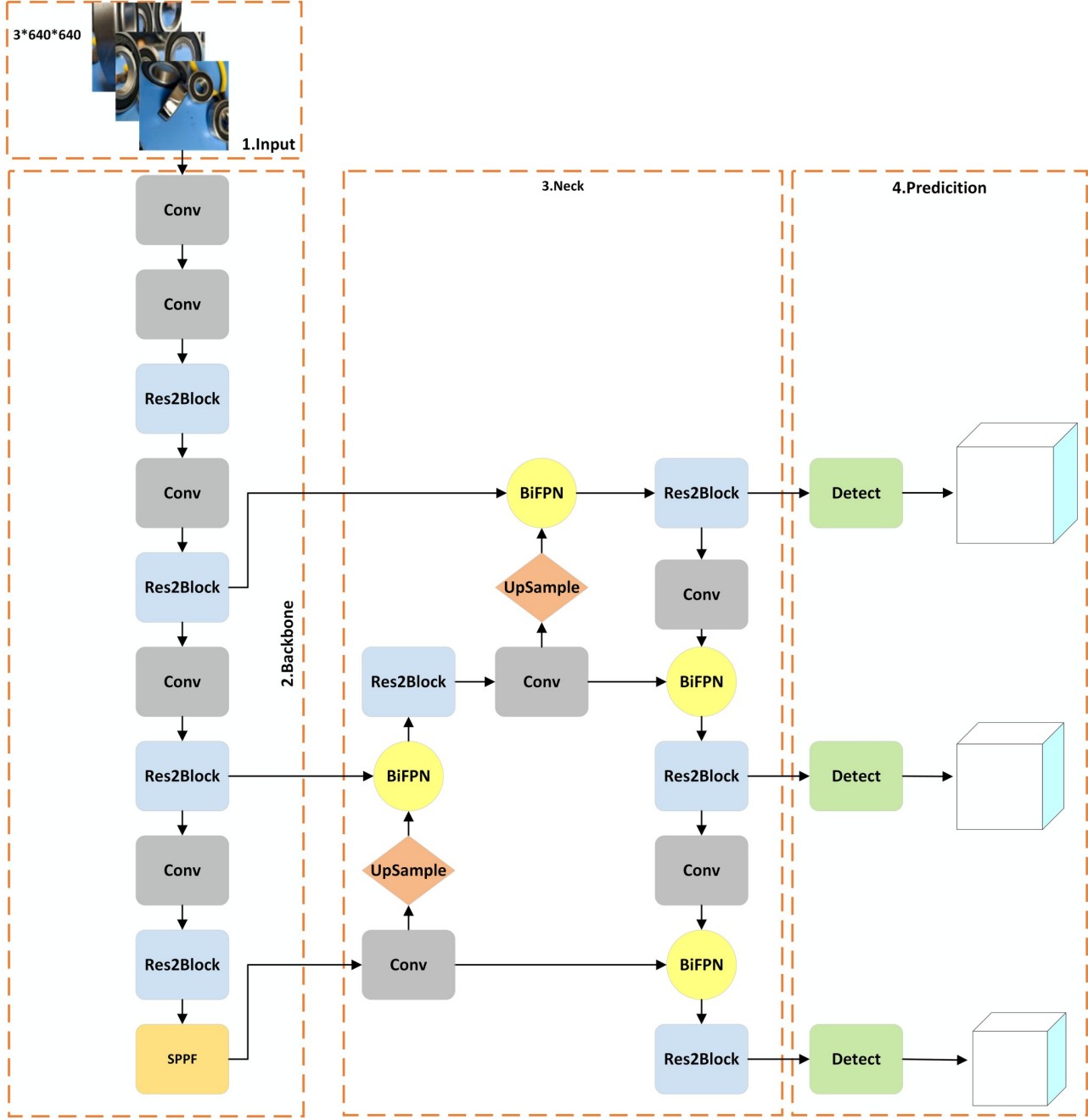

**Fig 2. Model structural of the improved YOLOv5 algorithm.**

2. Increasing the receptive field: The Res2Block module enhances the network's ability to capture a larger receptive field by adding multi-scale branches to the feature map. This helps the model better understand the contextual information within the input image, thereby effectively extracting features of the target object. This is beneficial for improving the recognition capability for small object defects.

The basic YOLOv5s algorithm utilizes a combination of Feature Pyramid Networks (FPN) and Path Aggregation Network (PAN) [17] to bolster the model's multi-scale feature fusion capabilities. Nevertheless, due to the fact that both FPN and PAN involve fusion operations of feature maps, some feature information is redundantly reused during the fusion process, leading to parameter redundancy. Furthermore, in scenarios where the image contains numerous densely packed targets, this type of feature fusion structure tends to generate a substantial number of overlapping prediction boxes, thereby escalating the complexity of non-maximum suppression (NMS). To solve this problem, the BiFPN [18] module is adopted. This module enables cross-fusion of features between different levels, allowing for the integration of shallow graphic features with deep semantic features while retaining the original features of each layer. This further enhances the model's learning efficiency and the accuracy of bearing defect identification.

## Re2Block module

The improved YOLOv5 algorithm described in this paper substitutes the CSPDarknet53 module with the Res2Block module for feature extraction, as illustrated in Fig 3.

As shown in Fig 3, the Res2Block module, following a $1 \times 1$ convolution, evenly divides the feature map into S subsets of feature maps, represented as $X_i$, where i $\in$ {1,2,...,S}. Compared to the input feature map, each feature subset $X_i$ has the same spatial size but the number of channels is 1/s [19]. Except for $X_1$, each $X_i$ is associated with a $3 \times 3$ convolution, denoted as $K_i$. Let $Y_i$ represent the output of $K_i$. The feature subset $X_i$ is added to the output of $K_i - 1$, and then fed into $K_i$ [20]. Finally, the feature maps of all groups are concatenated together and passed through another set of $1 \times 1$ convolutions to fully fuse the information. The definition of $Y_i$ is given by Eq (1)

$$Y_i = \begin{cases} X_i & i = 1 \\ K_i(X_i + Y_{i-1}) & 1 < i \leq s \end{cases} \tag{1}$$

The Res2Block module can fuse information from multiple branches within a single residual block to enhance the network's capability to express features at different resolutions. It can generate combinations of receptive fields at a finer granularity, bringing more information gain. Therefore, the Res2Block module is able to improve the recognition ability for small target defects in bearings and handle more complex recognition tasks.

## BiFPN module

The improved YOLOv5 algorithm discussed in this paper adopts the BiFPN module for feature fusion, replacing the PANet. The structure of BiFPN module is shown in Fig 4.

As shown in Fig 4, the BiFPN module first applies a convolution operation with a kernel of $3 \times 3$ and a stride of 1 to the $80 \times 80$ feature map, thereby obtaining a $40 \times 40$ feature map. This is then fused with the original $40 \times 40$ feature information. The fused $40 \times 40$ feature map undergoes the same convolution operation to produce a $20 \times 20$ feature map, which is then fused with the original $20 \times 20$ feature map. Subsequently, the total fused information is upsampled by a factor of 2, and finally, the three different sizes of feature maps are fused together. Therefore, the BiFPN module enables the full integration of shallow graphic features with deep semantic features, making full use of the information obtained during the feature extraction step. This helps to improve the model's training accuracy and reduce the rate of missed detections.

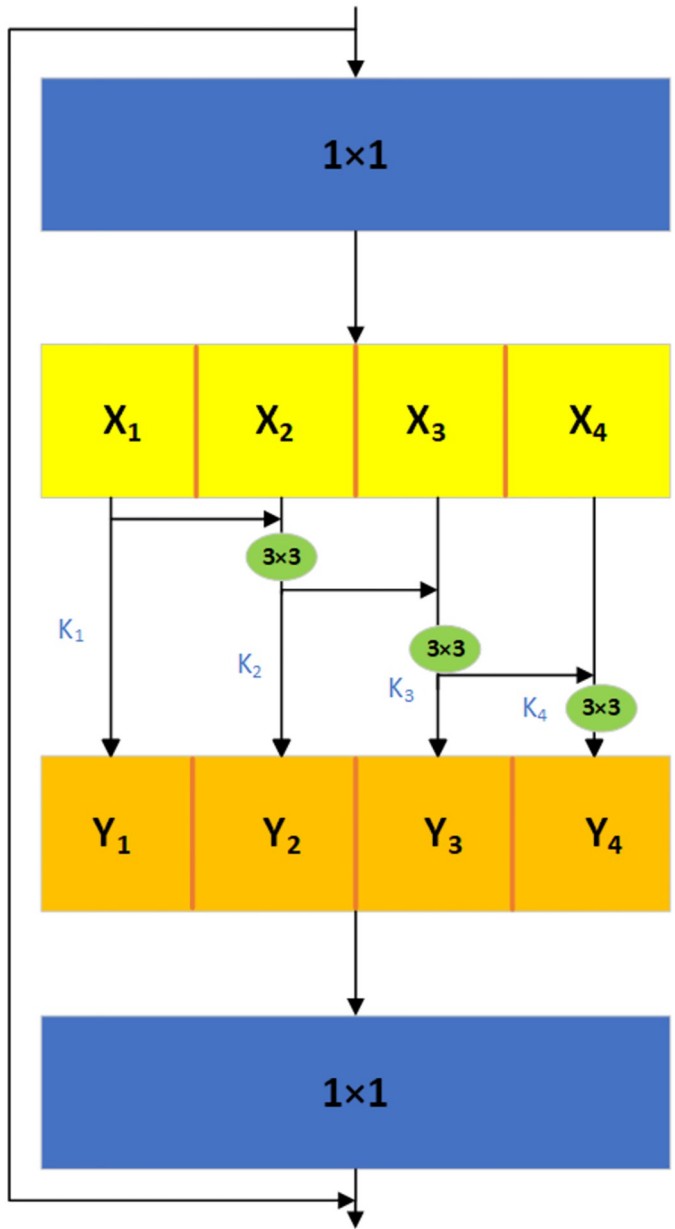

**Fig 3. The structure of the Res2Block module.**

## Pre-experiment preparation

### Dataset

The bearing defect dataset used in this paper was independently collected and contains three types of surface defects on bearings: "cashang" means "scratch", "huahen" means "scratch mark", and "aocao" means "groove". The dataset comprised 5,824 images, each image in the dataset contains multiple bearing defects. The defects on the bearing images were labeled using the LabelImg software, and corresponding txt format files were generated, which include information on the defect locations and categories. The dataset was divided into training,

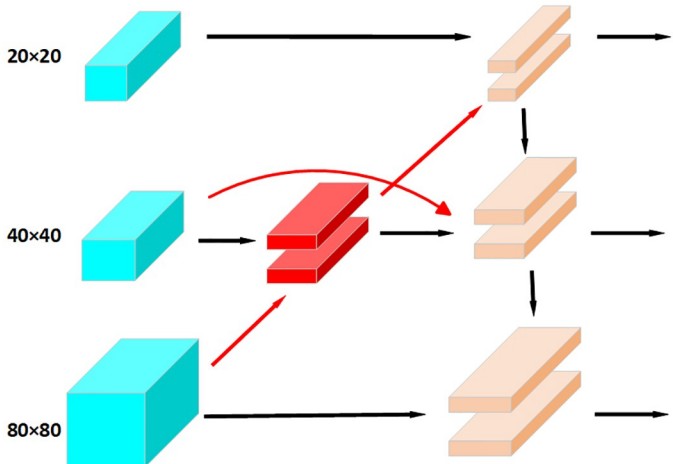

**Fig 4. The structure of the BiFPN.**

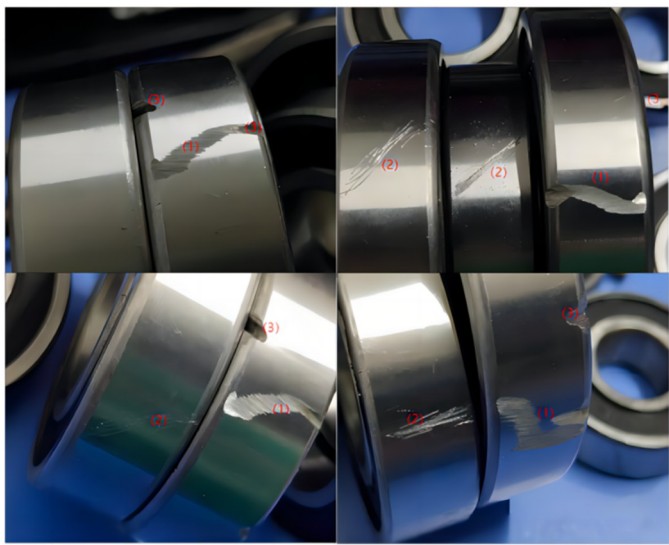

**Fig 5. Bearing defect dataset.** In the picture, (1) "cashang" means "scratch" (2) "huahen" means "scratch mark" (3) "aocao" means "groove".

validation, and test sets at a ratio of 8:1:1. The training set is used to train the model; the validation set is used to adjust the hyperparameters of the model and monitor the performance of the model; the test set is used to ultimately evaluate the performance of the model. The bearing defect dataset is shown in Fig 5.

## Data augmentation

To fully utilize the feature extraction, detection, and classification capabilities of the improved YOLOv5 algorithm presented in this paper, the dataset was initially expanded from 1,841 images to 5,824 by randomly adding Gaussian white noise as a method of dataset augmentation. Subsequently, during the training process, the Mosaic data augmentation method was

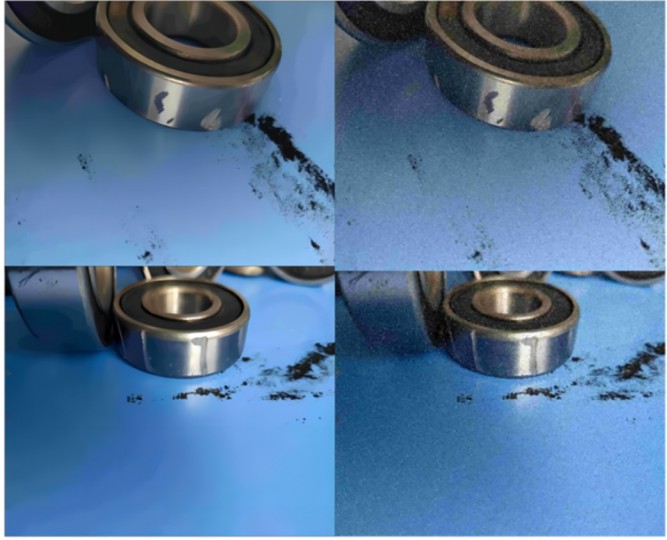

**Fig 6. Randomly adding Gaussian white noise.**

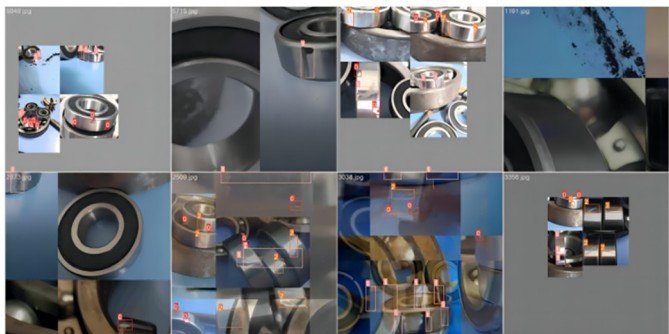

**Fig 7. Mosaic dataset augmentation.**

employed to further enrich the dataset, thereby enhancing the robustness and generalization ability of the algorithm model. The dataset augmentation methods are illustrated in Figs 6 and 7, respectively.

As illustrated in Fig 7, the Mosaic data augmentation method works as follows: First, four images are randomly selected from the training dataset, and these selected images are stitched together in a random order to form one large image. Next, a region is randomly chosen within the large image, ensuring that this region contains a target object, and this area is then cropped. Finally, the cropped area undergoes random scaling and translation operations to simulate variations of the target object at different scales and positions. This method increases the number of small samples in the dataset, thereby enhancing the algorithm's robustness.

## Experimental environment

The experimental platform for this study operates on Windows 10 Professional edition, with an NVIDIA GeForce RTX 3060 graphics card featuring 12GB of VRAM, and the system has

64GB of RAM. GPU acceleration is achieved through CUDA 12.0. Using this platform, training was conducted for the basic YOLOv5s algorithm, Small_obj algorithm, and the improved algorithm based on YOLOv5.

## Experimental evaluation indicators

In this experiment, the evaluation metrics include Precision (P), Recall (R), mean Average Precision (mAP), and Frames-per-second (FPS). P refers to the proportion of positive identifications that were correctly predicted among all predicted as positive. R indicates the proportion of actual positive cases that were correctly identified among all actual positives. mAP represents the average precision for each class. FPS denotes the speed at which the model processes images or videos, the higher the FPS, the faster the processing speed. The calculation formulas for P, R, and mAP are as shown in Eqs (2), (3), (4) and (5) respectively.

$$P = \frac{TP}{TP + FP} \tag{2}$$

$$R = \frac{TP}{TP + FN} \tag{3}$$

$$mAP = \frac{1}{|N|} \sum_{i=1}^{n} AP_i \tag{4}$$

$$FPS = \frac{frameNum}{elapsedTime} \tag{5}$$

In Eqs (2) and (3), TP, FP, and FN represent the number of true positives (correctly identified positive samples), false positives (incorrectly classified as positive samples), and false negatives (incorrectly classified as negative samples), respectively. In Eq (4), AP refers to the area under the P-R (Precision-Recall) curve, and mAP is obtained by averaging the AP across all defect categories. $AP_i$ denotes the average precision for the i-th type of defect, and n represents the number of defect types. In Eq (5), frameNum represents the number of frames recorded within a fixed period of time, elapsedTime represents the time that has passed.

## Relevant experimental parameters

1. At the stage of model training, the img_size is 640 pixels, the epoch is set to 100 epochs, the optimizer selected is Stochastic Gradient Descent (SGD), the volume of images input at one time (batch size) is set to 8, and the label_smoothing is set to 0.1.

2. During the model validation stage, the volume of images input at one time (batch size) is set to 8, and both the confidence threshold and the Non-Maximum Suppression Intersection over Union Threshold (NMS IoU threshold) are set to 0.5.

3. In the stage of bearing defect detection, the img_size is 640 pixels, and both the confidence threshold and Non-Maximum Suppression Intersection over Union Threshold (NMS IoU threshold) are set to 0.5.

## Experimental results and discussion

### Data loss

At the initial stage of model training, the epoch is set to 100 epochs, the optimizer selected is Stochastic Gradient Descent (SGD), and the volume of images input at one time (batch size) is set to 8. Upon completion of the model training, three types of loss curves can be obtained, which are related to the training process: bounding box coordinate loss (box_loss), object confidence loss (obj_loss), and classification loss (cls_loss). Regarding these three types of loss curves: The smaller the box_loss, the smaller the error between the predicted box and the ground truth box, indicating that the algorithm can accurately predict the target's position. The smaller the object_loss, the more accurately the algorithm can identify targets in the images. The smaller the cls_loss, the more accurate the algorithm is at classifying defect targets. The loss curve is shown in Fig 8.

From the loss curves shown in Fig 8, it's evident that the improved YOLOv5 algorithm (Ours) compared to the other two algorithms, has all three types of loss curves positioned at the lowest level, during the training processes. This indicates that the improved YOLOv5 algorithm can more accurately detect bearing defects and their positions, and can effectively classify them. Comparing the loss curves of the improved YOLOv5 algorithm and the Small_obj algorithm, it is clear that the improved YOLOv5 algorithm significantly reduces data loss during the training process. Therefore, the improved YOLOv5 algorithm has better performance in detecting small target defects.

### Model performance

The detection performance of the YOLOv5s algorithm, the Small_obj algorithm, and improved YOLOv5 algorithm (Ours) is shown in Table 1. In the table, "Cs" means "scratch", "Hh" means "scratch mark", "Ac" means "groove" and "All" means "average mAP". The units of P, R, and mAP are %, and the unit of FPS is frame/s. Compared to the YOLOv5s algorithm and the Small_obj algorithm, ours algorithm has achieved an increase of 7.2 and 4.9 percentage points in All, respectively. Moreover, the improved YOLOv5 algorithm achieves an mAP value of 0.939 for groove-type defects when detecting bearing defects. However, due to the model's complexity, there is a decrease in FPS, but it still meets the requirements for real-time detection. The experimental results indicate that the improved YOLOv5 algorithm not only has

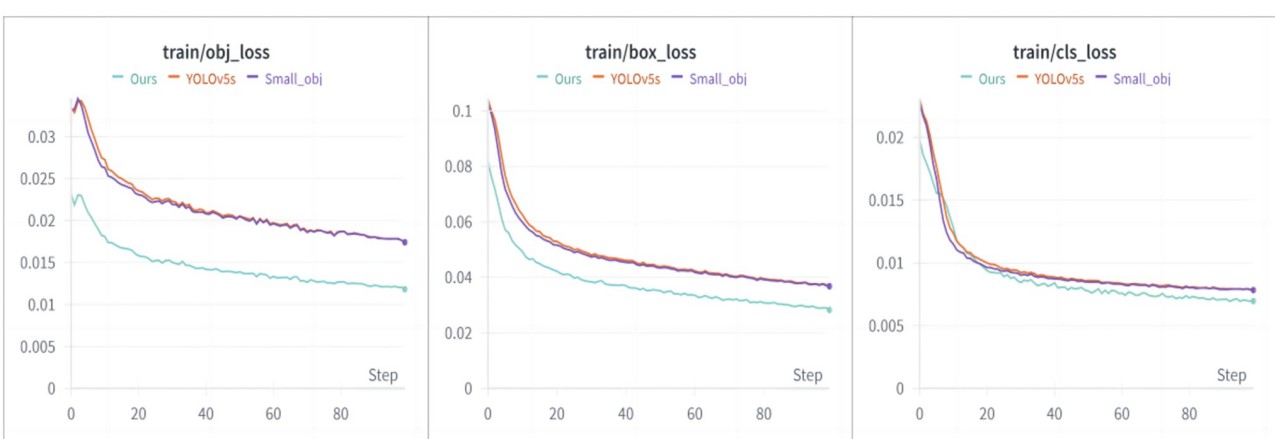

**Fig 8. Three types of loss curves.**

**Table 1. Algorithm detection performance.**

| Methods | P(Cs) | P(Hh) | P(Ac) | R(Cs) | R(Hh) | R(Ac) | mAP(Cs) | mAP(Hh) | mAP(Ac) | All | FPS |
|---------|-------|-------|-------|-------|-------|-------|---------|---------|---------|-----|-----|
| YOLOv5s | 0.929 | 0.826 | 0.934 | 0.753 | 0.736 | 0.746 | 0.844 | 0.785 | 0.807 | 0.812 | 84.493 |
| Small_obj | 0.924 | 0.847 | 0.928 | 0.805 | 0.794 | 0.759 | 0.837 | 0.803 | 0.865 | 0.835 | 66.382 |
| Ours | 0.951 | 0.863 | 0.957 | 0.835 | 0.842 | 0.899 | 0.887 | 0.825 | 0.939 | 0.884 | 57.656 |

higher recognition accuracy in the process of bearing defect detection but also satisfies the requirements for real-time detection.

## Detetion result

Random images were selected from the bearing defect dataset, and bearing defect detection experiments were conducted using the YOLOv5s algorithm, the Small_obj algorithm, and the improved YOLOv5 algorithm presented in this paper.

The detection results are shown in Fig 9.

The detection results shown in Fig 9 indicate that the YOLOv5 algorithm generally experiences missed detections during bearing defect detection, which is fatal for defect detection. Although the Small_obj algorithm's detection confidence is not significantly different from that of the improved YOLOv5 algorithm presented in this paper, the Small_obj algorithm still experiences missed detections in scenarios with dense or overlapping targets, and it even incorrectly identifies the types of bearing defects. Therefore, the improved YOLOv5 algorithm exhibits higher accuracy and robustness in bearing defect detection, meeting the requirements for practical bearing defect detection tasks.

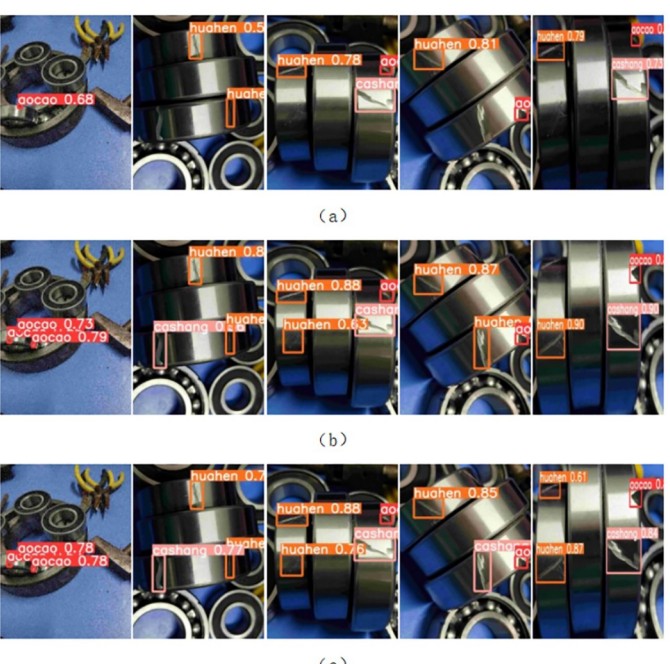

**Fig 9. Results of the bearing defect detection.** (a) The detection results of the YOLOv5 algorithm. (b) The detection results of the Small_obj algorithm. (c) The detection results of the improved YOLOv5 algorithm.

**Table 2. Ablation study on the improved YOLOv5 algorithm.**

| Method | P(cashang)(%) | P(huahen)(%) | P(aocao)(%) | mAP(%) | FPS (frame/s) |
|---|---|---|---|---|---|
| YOLOv5s | 0.929 | 0.826 | 0.934 | 0.812 | 74.493 |
| YOLOv5s+BiFPN | 0.938 | 0.858 | 0.942 | 0.873 | 61.588 |
| YOLOv5s+Res2Block | 0.942 | 0.849 | 0.948 | 0.869 | 56.382 |
| YOLOv5s+All | 0.951 | 0.863 | 0.955 | 0.884 | 57.656 |

**Table 3. Different algorithms for bearing defect detection performance.**

| Type | YOLOv3s | YOLOv5s | YOLOv6s | YOLOv7-tiny | YOLOv8 | YOLOv9s | Faster R-CNN | Ours |
|---|---|---|---|---|---|---|---|---|
| P(cashang)(%) | 0.782 | 0.929 | 0.931 | 0.933 | 0.936 | 0.953 | 0.812 | 0.951 |
| p(huahen)(%) | 0.738 | 0.826 | 0.839 | 0.842 | 0.847 | 0.859 | 0.762 | 0.863 |
| p(aocao)(%) | 0.819 | 0.934 | 0.937 | 0.939 | 0.941 | 0.953 | 0.859 | 0.957 |
| mAP(%) | 0.729 | 0.812 | 0.834 | 0.856 | 0.869 | 0.880 | 0.783 | 0.884 |
| FPS (frame/s) | 80.326 | 74.493 | 68.385 | 65.585 | 60.479 | 50.849 | 64.428 | 57.656 |

## Ablation experiment

Ablation experiments were set up to train and validate various algorithm improvement strategies separately. The precision for the three types of bearing defects, the mAP, and the FPS results are shown in Table 2. In the table, "cashang" means "scratch", "huahen" means "scratch mark", "aocao" means "groove" and "YOLOv5s+All" means "YOLOv5s+BiFPN+Res2Block".

The experimental data from Table 3 show that by adopting the Res2Block module in the feature extraction part, there has been a significant improvement in the accuracy of defect detection. Additionally, there has been a considerable improvement in the recognition of less conspicuous bearing defects, such as cashang (scratch).

In the feature fusion part, by adopting the BiFPN module, the mAP increased by 6.1 percentage points compared to the YOLOv5s algorithm. This indicates that the BiFPN module can better perform feature fusion, enhancing the model's ability to understand targets.

The improved YOLOv5 algorithm presented in this paper shows an increase in the recognition accuracy for all three types of defects compared to the YOLOv5s algorithm, with an mAP increase of 7.2 percentage points. However, due to the complexity of the algorithm model, the FPS decreased by 26.837 frames/s, but it still meets the requirements for real-time detection. Therefore, the experimental data in Table 3 validate the feasibility of the improved YOLOv5 algorithm.

## Comparative experiment

To further verify the detection performance of the improved YOLOv5 algorithm presented in this paper, a comparative experiment was conducted between the improved YOLOv5 algorithm(Ours) and other mainstream defect detection algorithms. The data in Table 3 compares the performance of different algorithms in terms of accuracy, mAP, and FPS. In the table, "cashang" means "scratch", "huahen" means "scratch mark", "aocao" means "groove".

The improved YOLOv5 algorithm has a higher mAP value than v3s, v5s, v6s, v7-tiny, v8, v9s and Faster R-CNN. In terms of FPS, due to the more complex model structure of the improved YOLOv5 algorithm, there is a varying degree of reduction in FPS compared to other algorithms. Nevertheless, it still meets the requirement for real-time detection.

## Conclusion

To address the challenges encountered in bearing defect detection, this paper presents improvements to the YOLOv5 algorithm. By leveraging enhanced capabilities for feature extraction and feature fusion, the proposed approach aims to increase the accuracy of bearing defect detection and reduce the rate of missed detections. Initially, the algorithm model's backbone was enhanced by incorporating the more finely grained Res2Block feature extraction module. This improvement expanded the model's receptive field, enabling it to better capture the subtle features within bearing images and thus improve its ability to recognize small target defects in bearings. Subsequently, we reconstructed the pyramid fusion network to enhance the integration capability of shallow image features with deep semantic features, thereby increasing the utilization rate of feature information. This enables the model to better understand the details within bearing images, further improving the accuracy of bearing defect recognition. At the same time, by randomly adding Gaussian white noise and mosaic techniques as data augmentation methods, we further enriched the dataset, enhancing the algorithm's robustness and generalization capability. Finally, the performance of the improved YOLOv5 algorithm is validated through ablation experiments and comparative experiments with other defect detection algorithms, including the Small_obj algorithm. The experimental results demonstrate that the improved YOLOv5 algorithm exhibits high mAP and accuracy in bearing defect detection, enabling more precise identification the types of small target defects on bearings in complex scenarios with multiple coexisting defects and overlapping detection targets, thereby providing valuable reference for practical bearing defect detection.

## Acknowledgments

I would like to express my heartfelt gratitude to all those who have provided support, encouragement, and assistance throughout this research process. First and foremost, I extend my sincere thanks to my advisor, Teacher Jiao Peigang, for your meticulous guidance and valuable advice, which have nurtured my academic growth and provided me with invaluable experience and inspiration. Additionally, I would like to thank my classmates, who have offered precious suggestions and encouragement during the writing of this thesis. Your friendship and support have brought warmth and fulfillment to both my academic and personal life.

## Author Contributions

**Conceptualization:** Weibo DU.

**Software:** Jiaming Ding.

**Validation:** Kangning Li.

**Writing – original draft:** Kangning Li.

**Writing – review & editing:** Kangning Li, Peigang Jiao.

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
