## [Decision Letter · Decision Letter 0]

17 Jun 2024

PONE-D-24-08858Bearing defect detection based on the improved YOLOv5 algorithmPLOS ONE

Dear Dr. Jiao,

Thank you for submitting your manuscript to PLOS ONE. After careful consideration, we feel that it has merit but does not fully meet PLOS ONE’s publication criteria as it currently stands. Therefore, we invite you to submit a revised version of the manuscript that addresses the points raised during the review process.

**ACADEMIC EDITOR: **As per the reviewers recommendation, I am suggesting "Major Revision". Authors need to go through each comments of all the Reviewers and make a point to point correction and upgrade the paper to be eligible for further consideration in "PLOS ONE". 

We look forward to receiving your revised manuscript.

Kind regards,

Himadri Majumder, Ph.D

Academic Editor

PLOS ONE

Journal Requirements:

3. Thank you for stating the following financial disclosure: "This work was supported in part by the Shandong Provincial Department of Science and Technology Research Project on Key Technologies for the Development of All-Terrain Intelligent Orchard Platform. (2019GNC106032)." 

Additional Editor Comments:

As per the reviewers recommendation, I am suggesting "Major Revision". Authors need to go through each comments of all the Reviewers and make a point to point correction and upgrade the paper to be eligible for further consideration in "PLOS ONE".

Reviewers' comments:

Reviewer's Responses to Questions

**Comments to the Author**

1. Is the manuscript technically sound, and do the data support the conclusions?

Reviewer #1: Partly

Reviewer #2: Partly

Reviewer #3: Partly

Reviewer #4: Partly

2. Has the statistical analysis been performed appropriately and rigorously? 

Reviewer #1: Yes

Reviewer #2: No

Reviewer #3: No

Reviewer #4: No

3. Have the authors made all data underlying the findings in their manuscript fully available?

Reviewer #1: Yes

Reviewer #2: Yes

Reviewer #3: No

Reviewer #4: Yes

4. Is the manuscript presented in an intelligible fashion and written in standard English?

Reviewer #1: Yes

Reviewer #2: No

Reviewer #3: No

Reviewer #4: Yes

5. Review Comments to the Author

Reviewer #1: A method of bearing defect detection based on an improved YOLOv5 algorithm is proposed to address the problem of low efficiency and easy omission in manual detection. The following problems are suggested to be improved.

1) There are problems in grammar which need to be improved.

2) This manuscript only compares with YOLOv5s. The proposed algorithm should compared with YOLOv7 and YOLOv8.

Reviewer #2: The paper is based on the YOLOv5 object detection algorithm, and achieves partial performance improvement by incorporating the Res2Block module and a Bidirectional Feature Pyramid Network. However, there are still some shortcomings in the content of the paper. My suggestions are as follows:

1.What does "all mAP" in the abstract mean? Does it refer to the average mAP across the dataset, or is it the mAP improvement for each defect class? mAP is a metric with various types; which type are you referring to? The contribution section in the introduction has the same issue.

2.The English writing needs improvement. For example, in the introduction, the sentence “Firstly, they enhanced the contrast of the images used, then applied the Hough algorithm to extract special areas, and finally performed edge template matching on the images to determine if there are defects”. Consider thoroughly revising the manuscript for language quality.

3.When convolutional neural network is first mentioned in the introduction, it should be followed by its abbreviation (CNN) in parentheses; subsequently, only the abbreviation should be used.

4.The first letter of a new sentence should be capitalized. Similar writing errors should be minimized throughout the manuscript.

5.In referencing Chinese authors, the surname should be used instead of the full name, as seen in references 2 and 14 in the introduction.

6.Defect names in the dataset should not be in Pinyin, such as "cashang, huahen, aocao." This applies to the tables of experimental results as well.

7.Comparing training and validation losses separately is not necessary. Consider using a table to compare the training losses between different networks.

8.When comparing model performance, specify which defect class achieves the stated mAP value, as in "the YOLOv5 algorithm achieves an mAP50 value of 0.939 when detecting bearing defects."

9.Consider merging Tables 1 and 2 into a single table. The term "All" in the table needs to be clearly defined in the manuscript.

10.Besides the proposed algorithm, there is only one comparative algorithm mentioned. It might be beneficial to include additional algorithms for comparison, such as YOLOv3, which you mentioned in the paper, to further validate the advantages of your proposed method in terms of inference speed and performance.

11.In the ablation study, clarify what "YOLOv5s+All" means. Ensure that this is clearly described in the manuscript and that the figures (or tables) and text are consistent.

Reviewer #3: To address the issues of low efficiency and easy omission of manual detection, the authors proposed an improved YOLOv5 algorithm to detect the bearing defect. After reading the whole paper, I think the contribution is weak, see the detailed comments as follows:

1. The innovation points of this article are general, where the proposed method is simply modified through YOLOv5.

2. Currently, the YOLO series has developed up to YOLOv10. Why is the author still using YOLOv5?

3. The literature review should be improved with the latest studies, and the research gap should be thoroughly investigated.

4. It should be clearly stated what the past researchers did, what their limitations were, and how the current study adds new knowledge to the existing literature.

5. The experimental section lacks comparative experiments with other state-of-the-art models, which makes it difficult to validate the effectiveness of the proposed method.

6. Language expression needs to be improved.

Reviewer #4: Overall the innovation and workload of this paper are insufficient in some degrees. It is expected that the author will carefully revise the suggestions provided in the future, improve the details of the experiment, supplement the comparative experiment, and reconsider the overall framework of the paper. It is also suggested to discuss more with other co-authors before submitting the revised draft to improve the quality of the paper. Below are some technical comments.

1、The overall format of the paper should be aligned at both ends, but the text format does align to the left.

2、Abstract part lacks the keyword extraction and writing.

3、In the paper review, some researchers have achieved good experimental results in the detection of bearing defects, and the detection result is as high as 98.6%. Therefore, they have doubts about the author's purpose of selecting the topic and carrying out the research. At the same time, the author sorts out the review and points out that some scholars' research methods have certain limitations, and these problems are not fully solved in the following paper.

4、The size of all the pictures in the paper is too large and in low resolution. For instance, in Fig1 & Fig2, the images are blurry and you can't even see the words in the red background. Please ask the authors to improve the clarity of the images and consider adjusting the font or color of the text to make the images legible.

5、Both Res2Block and BiFPN modules were proposed in 2019, and this year is 2024. Whether the improvement strategy has fallen behind, it is suggested that the author should contact or study some cutting-edge articles in the field of computer vision to ensure that the research process keeps pace with The Times.

6、The Neck part of the original YOLOv5 model adopted the feature pyramid (FPN) and path aggregation network (PANet) structures, while the author only described the PANet structure in the relevant chapters of the paper and used BiFPN module to replace the PANet structure, which indicates that the author has not a comprehensive understanding of the YOLOv5 network structure. It is suggested that the author revise and study the relevant articles carefully.

7、When introducing the dataset, the author has introduced the enhancement method of the dataset in detail. There are overlapping parts in this part, so it is suggested that the author delete the relevant chapters.

8、The author didn’t introduce the setting of relevant experimental parameters, so it is suggested that the author add this part.

9、The index of detection rate FPS is not given the full name in English and calculation formula.

10、The author should explicitly use the model of YOLOv5 version in the whole paper, such as YOLOv5s, YOLOv5m, YOLOv5l and YOLOv5x.

11、What is Small obj. algorithm? The improved algorithm is not compared with the original YOLOv5s algorithm in detail, but only provides the improved results of various indicators. It is suggested to supplement the experimental results of YOLOv5s in Table 1 and Table 2.

12、It is suggested that the author adjust the relevant chapters of comparison experiment and ablation experiment to ensure the overall logic of the article.

13、There are too few algorithms in the comparative experiment to demonstrate the superiority of the improved YOLOv5s algorithm in this paper. At the same time, in the literature review section, a scholar's detection result reached 98.6%, while the mAP of the algorithm proposed in this paper is 88.4%. We hope the author can provide stronger evidence to prove the superiority of the algorithm in this paper.

14、In Table1, Table 2 and Table 3, the names of the defects in the tables are in Chinese Pinyin, please replace the Chinese names with the corresponding English ones to avoid doubts from non-Chinese speakers. If necessary, explain the detected parts in the text in the form of words or pictures. For example, ‘cashang’ means ‘xxx (English Word)’

15、It is better to to add more textual information about the type of bearing damage in Fig 5

6. PLOS authors have the option to publish the peer review history of their article (what does this mean?). If published, this will include your full peer review and any attached files.

Reviewer #1: No

Reviewer #2: No

Reviewer #3: No

Reviewer #4: No

---

## [Author Response · Author response to Decision Letter 0]

10 Jul 2024

AUTHORS'RESPONSE TO EDITORS AND REVIEWERS

Dear Editors and Reviewers,

Thank you very much for your kindly comments on our manuscript. There is no doubt that these comments are valuable and very helpful for revising and improving our manuscript. We are pleased to submit the revised research article “Bearing defect detection based on the improved YOLOv5 algorithm”. On the following pages, we present our point-by-point responses to the comments.

We would like also to thank you for allowing us to resubmit a revised copy of the manuscript. We appreciate your time and look forward to your response.

Sincerely,

Peigang Jiao,Ph.D. (corresponding author)

jiaopeigang@163.com

Comments from Academic Editor

Thank you for submitting your manuscript to PLOS ONE. After careful consideration, we feel that it has merit but does not fully meet PLOS ONE’s publication criteria as it currently stands. Therefore, we invite you to submit a revised version of the manuscript that addresses the points raised during the review process.

A rebuttal letter that responds to each point raised by the academic editor and reviewer(s). You should upload this letter as a separate file labeled 'Response to Reviewers'.

A marked-up copy of your manuscript that highlights changes made to the original version. You should upload this as a separate file labeled 'Revised Manuscript with Track Changes'.

An unmarked version of your revised paper without tracked changes. You should upload this as a separate file labeled 'Manuscript'.

Response: Thank you for your kind guidance. We have resubmitted our revised manuscript as requested.

Response: Thank you for your helpful advice. We would like to change our financial disclosure and have updated the statement in the cover letter, while also resubmitting the image files in accordance with the guidelines.

If applicable, we recommend that you deposit your laboratory protocols in protocols.io to enhance the reproducibility of your results. Protocols.io assigns your protocol its own identifier (DOI) so that it can be cited independently in the future. For instructions see:

https://journals.plos.org/plosone/s/submission-guidelines#loc-laboratory-protocols. Additionally, PLOS ONE offers an option for publishing peer-reviewed Lab Protocol articles, which describe protocols hosted on protocols.io. Read more information on sharing protocols at https://plos.org/protocols?

utm_medium=editorial-

email&utm_source=authorletters&utm_campaign=protocols.

We look forward to receiving your revised manuscript.

Kind regards,

Journal Requirements

Response: Thank you for your kind guidance. We have reformatted the manuscript according to the above style guidelines.

Response: Thank you for your kind guidance. We have shared our code in the Figshare database. Doi: https://doi.org/10.6084/m9.figshare.25335076.v1;
https://doi.org/10.6084/m9.figshare.26197928.v1

3. Thank you for stating the following financial disclosure: "This work was supported in part by the Shandong Provincial Department of Science and Technology Research Project on Key Technologies for the Development of All-Terrain Intelligent Orchard Platform. (2019GNC106032)."

Response: Thank you for your kind guidance. We have already provided a clarification in the cover letter regarding the role of the funders in the study.

Additional Editor Comments:

As per the reviewers recommendation, I am suggesting "Major Revision". Authors need to go through each comments of all the Reviewers and make a point to point correction and upgrade the paper to be eligible for further consideration in "PLOS ONE".

Response: Thank you for your advice. We will address reviewers’ comments in the following pages.

Thank you again for your kind and helpful advice

AUTHOR’S RESPONSE TO REVIEWERS’ COMMENTS

We greatly appreciate the reviewers taking the time to provide constructive comments and helpful suggestions. There is no doubt that the suggestions have significantly raised the quality if the manuscript and have enable us to improve the manuscript. Each suggested revision and comment brought forward by the reviewers was accurately incorporated and considered. We have carefully addressed all the reviewer's concerns. Please see our replies. Changes highlighted in red have been made accordingly in the revised manuscript.

Comments to the Author

1. Is the manuscript technically sound, and do the data support the conclusions?

Reviewer #1: Partly

Reviewer #2: Partly

Reviewer #3: Partly

Reviewer #4: Partly

Response: Thanks for your review. We have further improved the contrast experiments, and finally verified the performance of the improved YOLOv5 algorithm in bearing defect detection through ablation experiments and contrast experiments.

2. Has the statistical analysis been performed appropriately and rigorously?

Reviewer #1: Yes

Reviewer #2: No

Reviewer #3: No

Reviewer #4: No

Response: Thank you for agreeing that our analysis is appropriate and rigorous.We will further improve the statistical analysis.

3. Have the authors made all data underlying the findings in their manuscript fully available?

Reviewer #1: Yes

Reviewer #2: Yes

Reviewer #3: No

Reviewer #4: Yes

Response: Thank you for your approval for our data.

4. Is the manuscript presented in an intelligible fashion and written in standard English?

Reviewer #1: Yes

Reviewer #2: No

Reviewer #3: No

Reviewer #4: Yes

Response: Thank you for your approval. We will carefully revise the manuscript for language quality.

Review Comments to the Author

Reviewer #1: A method of bearing defect detection based on an improved YOLOv5 algorithm is proposed to address the problem of low efficiency and easy omission in manual detection. The following problems are suggested to be improved.

1. There are problems in grammar which need to be improved.

Response: Thanks to the reviewer for pointing out the problem. We have improved the grammatical issues in the paper.

2. This manuscript only compares with YOLOv5s. The proposed algorithm should compared with YOLOv7 and YOLOv8.

Response: Thanks to the reviewer for pointing out the problem. We have augmented the "Comparison experiment" section by including a comparative analysis with other mainstream defect detection algorithms in terms of Precision (P), mean Average Precision (mAP), and Frames Per Second (FPS). Including a comparison with YOLOv7 and YOLOv8 algorithms.

Reviewer #2: The paper is based on the YOLOv5 object detection algorithm, and achieves partial performance improvement by incorporating the Res2Block module and a Bidirectional Feature Pyramid Network. However, there are still some shortcomings in the content of the paper. My suggestions are as follows:

1.What does "all mAP" in the abstract mean? Does it refer to the average mAP across the dataset, or is it the mAP improvement for each defect class? mAP is a metric with various types; which type are you referring to? The contribution section in the introduction has the same issue. 

Response: Thanks to the reviewer for pointing out the problem. In the abstract and introduction sections of the paper, "all mAP" indeed refers to the average mAP across the dataset. We clarified this in the revised manuscript using red font.

2.The English writing needs improvement. For example, in the introduction, the sentence “Firstly, they enhanced the contrast of the images used, then applied the Hough algorithm to extract special areas, and finally performed edge template matching on the images to determine if there are defects”. Consider thoroughly revising the manuscript for language quality.

Response: Thanks to the reviewer for pointing out the problem. We have revised the relevant sentence content and retranslated the manuscript to improve the language quality.

3.When convolutional neural network is first mentioned in the introduction, it should be followed by its abbreviation (CNN) in parentheses; subsequently, only the abbreviation should be used.

Response: Thanks to the reviewer for pointing out the problem. We have revised the relevant content in the "Introduction" chapter and highlighted it with red font in the manuscript.

4.The first letter of a new sentence should be capitalized. Similar writing errors should be minimized throughout the manuscript.

Response: Thanks to the reviewer for pointing out the problem. We have revised the relevant content and highlighted it with red font in the manuscript.

5.In referencing Chinese authors, the surname should be used instead of the full name, as seen in references 2 and 14 in the introduction.

Response: Thanks to the reviewer for pointing out the problem. The following example is the official reference format for Chinese authors in PLOS ONE, and we have modified the corresponding references in the manuscript according to this format and highlighted them in red font .

Example：Hou WR, Hou YL, Wu GF, Song Y, Su XL, Sun B, et al. cDNA, genomic sequence cloning and over expression of ribosomal protein gene L9 (rpL9) of the giant panda (Ailuropoda melanoleuca). Genet Mol Res. 2011;10: 1576-1588.

6.Defect names in the dataset should not be in Pinyin, such as "cashang, huahen, aocao." This applies to the tables of experimental results as well.

Response: Thanks to the reviewer for pointing out the problem. We have provided explanations for the names of relevant bearing defects and have marked the relevant content in red font in the "Dataset" section, "Model performance" section, "Ablation experiment" section and "Comparative experiment" section of the manuscript.

7.Comparing training and validation losses separately is not necessary. Consider using a table to compare the training losses between different networks.

Response: Thanks you for your advice. After carefully considering your suggestions, we realize that while using a table to compare the training losses of different networks is concise and clear, it may not visually illustrate the changing trend of the losses during the training process. Therefore, we have decided to continue using our original approach to display the loss changes of different models during training. Meanwhile, to maintain the conciseness and focus of the paper, we have decided to only retain the comparison of training losses and omit the comparison of data loss curves during the validation process.

8.When comparing model performance, specify which defect class achieves the stated mAP value, as in "the YOLOv5 algorithm achieves an mAP50 value of 0.939 when detecting bearing defects."

Response: Thanks to the reviewer for pointing out the problem. We have made revisions to the relevant content in the "Model performance" section and highlighted the related content in red font in the manuscript.

9.Consider merging Tables 1 and 2 into a single table. The term "All" in the table needs to be clearly defined in the manuscript.

Response: Thanks you for your advice. We have merged Table 1 and Table 2 into a new Table 1. The corresponding content in the "Model performance" section of the manuscript has been marked in red. In the table, "All" refers to the average mAP across the dataset.

10.Besides the proposed algorithm, there is only one comparative algorithm mentioned. It might be beneficial to include additional algorithms for comparison, such as YOLOv3, which you mentioned in the paper, to further validate the advantages of your proposed method in terms of inference speed and performance.

Response: Thanks to the reviewer for pointing out the problem. We have augmented the "Comparison experiment" section by including a comparative analysis with other mainstream defect detection algorithms in terms of Precision (P), mean Average Precision (mAP), and Frames Per Second (FPS). Including a comparison with YOLOv3 algorithms.

11.In the ablation study, clarify what "YOLOv5s+All" means. Ensure that this is clearly described in the manuscript and that the figures (or tables) and text are consistent.

Response: Thanks to the reviewer for pointing out the problem. "YOLOv5s+All" means "YOLOv5s+BiFPN+Res2Block". We have provided an explanation for the relevant content in the "Ablation experiment" section, and highlighted it in red font in the manuscript.

Reviewer #3: To address the issues of low efficiency and easy omission of manual detection, the authors proposed an improved YOLOv5 algorithm to detect the bearing defect. After reading the whole paper, I think the contribution is weak, see the detailed comments as follows:

1.The innovation points of this article are general, where the proposed method is simply modified through YOLOv5.

Response: Thanks to the reviewer for pointing out the problem. Although it may seem like we have merely modified the YOLO

---

## [Decision Letter · Decision Letter 1]

23 Jul 2024

PONE-D-24-08858R1Bearing defect detection based on the improved YOLOv5 algorithmPLOS ONE

Dear Dr. Jiao,

Thank you for submitting your manuscript to PLOS ONE. After careful consideration, we feel that it has merit but does not fully meet PLOS ONE’s publication criteria as it currently stands. Therefore, we invite you to submit a revised version of the manuscript that addresses the points raised during the review process.

**Dear authors, we have now received reviewers feedback regarding your revised paper. As per their feedback, authors have upgrade and improved the paper content based on the suggestions provided. However, one reviewer is not convinced regarding the innovation in this article and the reviewer thinks it is not sufficient. Another reviewer suggest, the author should add relevant comparative experiments to compare the current mainstream object detection algorithms to verify the superiority of the improved algorithm in the comparative experiment section and more convincing explanations need to be added. In view of their comments, I am recommending "Minor Revision" to address this flaws. Authors are requested to go through the comments and made sufficient improvement of their paper.**

We look forward to receiving your revised manuscript.

Kind regards,

Himadri Majumder, Ph.D

Academic Editor

PLOS ONE

Journal Requirements:

Additional Editor Comments:

Dear authors, we have now received reviewers feedback regarding your revised paper. As per their feedback, authors have upgrade and improved the paper content based on the suggestions provided. However, one reviewer is not convinced regarding the innovation in this article and the reviewer thinks it is not sufficient. Another reviewer suggest, the author should add relevant comparative experiments to compare the current mainstream object detection algorithms to verify the superiority of the improved algorithm in the comparative experiment section and more convincing explanations need to be added. In view of their comments, I am recommending "Minor Revision" to address this flaws. Authors are requested to go through the comments and made sufficient improvement of their paper.

Reviewers' comments:

Reviewer's Responses to Questions

**Comments to the Author**

1. If the authors have adequately addressed your comments raised in a previous round of review and you feel that this manuscript is now acceptable for publication, you may indicate that here to bypass the “Comments to the Author” section, enter your conflict of interest statement in the “Confidential to Editor” section, and submit your "Accept" recommendation.

Reviewer #1: All comments have been addressed

Reviewer #2: (No Response)

Reviewer #3: All comments have been addressed

Reviewer #4: All comments have been addressed

2. Is the manuscript technically sound, and do the data support the conclusions?

Reviewer #1: Yes

Reviewer #2: Partly

Reviewer #3: Yes

Reviewer #4: Partly

3. Has the statistical analysis been performed appropriately and rigorously? 

Reviewer #1: Yes

Reviewer #2: Yes

Reviewer #3: Yes

Reviewer #4: Yes

4. Have the authors made all data underlying the findings in their manuscript fully available?

Reviewer #1: Yes

Reviewer #2: Yes

Reviewer #3: Yes

Reviewer #4: Yes

5. Is the manuscript presented in an intelligible fashion and written in standard English?

Reviewer #1: Yes

Reviewer #2: No

Reviewer #3: Yes

Reviewer #4: Yes

6. Review Comments to the Author

**Reviewer #1: **(No Response)

**Reviewer #2:** The author has made some improvements to the content based on the suggestions provided. However, the innovation in this article is not sufficient.

**Reviewer #3:** (No Response)

**Reviewer #4:** After careful modification by the author, the overall quality of this paper has been further improved. The topic selection of this paper has certain research significance, the structure and logic of the paper are clear, the language expression is standardized, and the research results have certain construction significance.

Finally, it is suggested that the author try to pay attention to and contact with some current mainstream and newly launched object detection algorithms. Improve the comparison experiment part, improve the persuasiveness and superiority of the improved algorithm proposed in this paper.

7. PLOS authors have the option to publish the peer review history of their article (what does this mean?). If published, this will include your full peer review and any attached files.

Reviewer #1: No

Reviewer #2: No

Reviewer #3: No

Reviewer #4: No

---

## [Author Response · Author response to Decision Letter 1]

2 Aug 2024

AUTHORS'RESPONSE TO EDITORS AND REVIEWERS

Dear Editors and Reviewers,

Thank you very much for your kindly comments on our manuscript. There is no doubt that these comments are valuable and very helpful for revising and improving our manuscript. We are pleased to submit the revised research article “Bearing defect detection based on the improved YOLOv5 algorithm”. On the following pages, we present our point-by-point responses to the comments.

We would like also to thank you for allowing us to resubmit a revised copy of the manuscript. We appreciate your time and look forward to your response.

Sincerely,

Peigang Jiao,Ph.D. (corresponding author)

jiaopeigang@163.com

Comments from Academic Editor

Thank you for submitting your manuscript to PLOS ONE. After careful consideration, we feel that it has merit but does not fully meet PLOS ONE’s publication criteria as it currently stands. Therefore, we invite you to submit a revised version of the manuscript that addresses the points raised during the review process.

Dear authors, we have now received reviewers feedback regarding your revised paper. As per their feedback, authors have upgrade and improved the paper content based on the suggestions provided. However, one reviewer is not convinced regarding the innovation in this article and the reviewer thinks it is not sufficient. Another reviewer suggest, the author should add relevant comparative experiments to compare the current mainstream object detection algorithms to verify the superiority of the improved algorithm in the comparative experiment section and more convincing explanations need to be added. In view of their comments, I am recommending "Minor Revision" to address this flaws. Authors are requested to go through the comments and made sufficient improvement of their paper.

A rebuttal letter that responds to each point raised by the academic editor and reviewer(s). You should upload this letter as a separate file labeled 'Response to Reviewers'.

A marked-up copy of your manuscript that highlights changes made to the original version. You should upload this as a separate file labeled 'Revised Manuscript with Track Changes'.

An unmarked version of your revised paper without tracked changes. You should upload this as a separate file labeled 'Manuscript'.

Response: Thank you for your kind guidance. We have resubmitted our revised manuscript as requested.

Response: Thank you for your helpful advice. We don't need to revise the financial disclosure.

If applicable, we recommend that you deposit your laboratory protocols in protocols.io to enhance the reproducibility of your results. Protocols.io assigns your protocol its own identifier (DOI) so that it can be cited independently in the future. For instructions see:

https://journals.plos.org/plosone/s/submission-guidelines#loc-laboratory-protocols. Additionally, PLOS ONE offers an option for publishing peer-reviewed Lab Protocol articles, which describe protocols hosted on protocols.io. Read more information on sharing protocols at https://plos.org/protocols?

utm_medium=editorial-email&utm_source=authorletters&utm_campaign=protocols.

We look forward to receiving your revised manuscript.

Kind regards,

Journal Requirements

Response: Thank you for your kind guidance. We have checked the reference list to ensure that it is complete and correct.

Additional Editor Comments:

Dear authors, we have now received reviewers feedback regarding your revised paper. As per their feedback, authors have upgrade and improved the paper content based on the suggestions provided. However, one reviewer is not convinced regarding the innovation in this article and the reviewer thinks it is not sufficient. Another reviewer suggest, the author should add relevant comparative experiments to compare the current mainstream object detection algorithms to verify the superiority of the improved algorithm in the comparative experiment section and more convincing explanations need to be added. In view of their comments, I am recommending "Minor Revision" to address this flaws. Authors are requested to go through the comments and made sufficient improvement of their paper.

Response: Thank you for your advice. We will address reviewers’ comments in the following pages.

AUTHOR’S RESPONSE TO REVIEWERS’ COMMENTS

We greatly appreciate the reviewers taking the time to provide constructive comments and helpful suggestions. There is no doubt that the suggestions have significantly raised the quality if the manuscript and have enable us to improve the manuscript. Each suggested revision and comment brought forward by the reviewers was accurately incorporated and considered. We have carefully addressed all the reviewer's concerns. Please see our replies. Changes highlighted in red have been made accordingly in the revised manuscript.

Comments to the Author

1. If the authors have adequately addressed your comments raised in a previous round of review and you feel that this manuscript is now acceptable for publication, you may indicate that here to bypass the “Comments to the Author” section, enter your conflict of interest statement in the “Confidential to Editor” section, and submit your "Accept" recommendation.

Reviewer #1: All comments have been addressed

Reviewer #2: (No Response)

Reviewer #3: All comments have been addressed

Reviewer #4: All comments have been addressed

Response: Thank you for your approval.

2. Is the manuscript technically sound, and do the data support the conclusions?

Reviewer #1: Yes

Reviewer #2: Partly

Reviewer #3: Yes

Reviewer #4: Partly

Response: Thanks for your review. We have further improved the contrast experiments, and finally verified the performance of the improved YOLOv5 algorithm in bearing defect detection through ablation experiments and contrast experiments.

3. Has the statistical analysis been performed appropriately and rigorously?

Reviewer #1: Yes

Reviewer #2: Yes

Reviewer #3: Yes

Reviewer #4: Yes

Response: Thank you for agreeing that our analysis is appropriate and rigorous.

4. Have the authors made all data underlying the findings in their manuscript fully available?

Reviewer #1: Yes

Reviewer #2: Yes

Reviewer #3: Yes

Reviewer #4: Yes

Response: Thank you for your approval for our data.

5. Is the manuscript presented in an intelligible fashion and written in standard English?

Reviewer #1: Yes

Reviewer #2: No

Reviewer #3: Yes

Reviewer #4: Yes

Response: Thank you for your approval. We will carefully revise the manuscript for language quality.

Review Comments to the Author

Reviewer #1: (No Response)

Reviewer #2: The author has made some improvements to the content based on the suggestions provided. However, the innovation in this article is not sufficient.

Response: Thanks to the reviewer for pointing out the problem. We modifications to the YOLOv5 algorithm are targeted and well-considered. We optimized and improved several crucial parts of YOLOv5 specifically addressing the unique challenges and difficulties of bearing defect detection tasks. For instance, we refined the network architecture to enhance the detection capability for minute bearing defects; optimized the feature extraction ability, allowing it to capture finer details in images and thus improving the recognition of small target defects in bearings; and enhanced the feature fusion capacity, effectively integrating shallow image features with deep semantic features, resulting in a reduction in missed detections and an increase in the accuracy of bearing defect recognition. These improvements are not mere superficial modifications; they have been thoroughly researched and experimentally validated.

To validate the effectiveness of our approach, we conducted experiments on dataset and compared it with the original YOLOv5 and other relevant defect detection methods. The results demonstrated that our method achieved superior performance in bearing defect detection tasks, not only improving accuracy but also exhibiting robustness in scenarios with dense and overlapping targets. These experimental outcomes strongly confirm that our method is not just a simple modification of YOLOv5 but possesses both practical significance and innovation.

Based on your valuable suggestions, we will further delve into the research and exploration of new methods and technologies for bearing defect detection. We will continue to keep abreast of the latest research achievements and technological advancements in this field, and strive to incorporate more innovative elements into our research.

Reviewer #3: (No Response)

Reviewer #4: After careful modification by the author, the overall quality of this paper has been further improved. The topic selection of this paper has certain research significance, the structure and logic of the paper are clear, the language expression is standardized, and the research results have certain construction significance.

Finally, it is suggested that the author try to pay attention to and contact with some current mainstream and newly launched object detection algorithms. Improve the comparison experiment part, improve the persuasiveness and superiority of the improved algorithm proposed in this paper.

Response: Thanks you for your advice. We have improved the comparison experiment part and highlighted it in red in the manuscript.

7. PLOS authors have the option to publish the peer review history of their article (what does this mean?). If published, this will include your full peer review and any attached files.

Do you want your identity to be public for this peer review? For information about this choice, including consent withdrawal, please see our Privacy Policy.

Reviewer #1: No

Reviewer #2: No

Reviewer #3: No

Reviewer #4: No

While revising your submission, please upload your figure files to the Preflight Analysis and Conversion Engine (PACE) digital diagnostic tool, https://pacev2.apexcovantage.com/. PACE helps ensure that figures meet PLOS requirements. To use PACE, you must first register as a user. Registration is free. Then, login and navigate to the UPLOAD tab, where you will find detailed instructions on how to use the tool. If you encounter any issues or have any questions when using PACE, please email PLOS at <a href="mailto:figures@plos.org">figures@plos.org. Please note that Supporting Information files do not need this step.

---

## [Decision Letter · Decision Letter 2]

23 Aug 2024

Bearing defect detection based on the improved YOLOv5 algorithm

PONE-D-24-08858R2

Dear Dr. Jiao,

We’re pleased to inform you that your manuscript has been judged scientifically suitable for publication and will be formally accepted for publication once it meets all outstanding technical requirements.

Kind regards,

Himadri Majumder, Ph.D

Academic Editor

PLOS ONE

Additional Editor Comments (optional):

As per reviewers feedback, I am happy to recommend "Acceptance" to this revised paper.

Reviewers' comments:

Reviewer's Responses to Questions

**Comments to the Author**

1. If the authors have adequately addressed your comments raised in a previous round of review and you feel that this manuscript is now acceptable for publication, you may indicate that here to bypass the “Comments to the Author” section, enter your conflict of interest statement in the “Confidential to Editor” section, and submit your "Accept" recommendation.

Reviewer #1: All comments have been addressed

Reviewer #4: All comments have been addressed

2. Is the manuscript technically sound, and do the data support the conclusions?

Reviewer #1: Yes

Reviewer #4: Yes

3. Has the statistical analysis been performed appropriately and rigorously? 

Reviewer #1: Yes

Reviewer #4: Yes

4. Have the authors made all data underlying the findings in their manuscript fully available?

Reviewer #1: Yes

Reviewer #4: Yes

5. Is the manuscript presented in an intelligible fashion and written in standard English?

Reviewer #1: Yes

Reviewer #4: Yes

6. Review Comments to the Author

Reviewer #1: (No Response)

Reviewer #4: The revised paper is much improved in quality. I am satisfied with the current version, so suggest acceptance.

7. PLOS authors have the option to publish the peer review history of their article (what does this mean?). If published, this will include your full peer review and any attached files.

Reviewer #1: No

Reviewer #4: No
